# Antibacterial Effect of Sesame Protein-Derived Peptides against *Escherichia coli* and *Staphylococcus aureus*: In Silico and In Vitro Analysis

**DOI:** 10.3390/nu16010175

**Published:** 2024-01-04

**Authors:** Zehui Zhu, Fei Pan, Ou Wang, Liang Zhao, Lei Zhao

**Affiliations:** 1Beijing Engineering and Technology Research Center of Food Additives, Beijing Technology and Business University, Beijing 100048, China; zhuzehui0614@163.com; 2Institute of Apicultural Research, Chinese Academy of Agricultural Sciences, Beijing 100093, China; yunitcon@yeah.net; 3National Institute for Nutrition and Health, Chinese Center for Disease Control and Prevention, Beijing 100050, China; wangou@ninh.chinacdc.cn

**Keywords:** sesame protein, antibacterial peptides, dihydropteroate synthase, molecular docking, *Escherichia coli*, *Staphylococcus aureus*

## Abstract

This study aimed to screen out antibacterial peptides derived from sesame (*Sesamum indicum* L.) through in silico and in vitro methods. In silico proteolysis of sesame proteins with pepsin, trypsin, and chymotrypsin was performed with the online server BIOPEP-UWM. The CAMPR3 online server was used to predict the antimicrobial effect of peptides. The ToxinPred, PepCalc, and AllergenFP tools were utilized to forecast the physicochemical properties, toxicity, and allergen of the peptides. Molecular docking analysis showed that six cationic antimicrobial peptides could directly interact with the key sites of dihydropteroate synthase, whereas Ala-Gly-Gly-Val-Pro-Arg and Ser-Thr-Ile-Arg exhibited the strongest binding affinity. In vitro antibacterial experiment showed the minimum inhibitory concentration (MIC) of Ser-Thr-Ile-Arg against *Escherichia coli* and *Staphylococcus aureus* was 1024 and 512 µg/mL, respectively. Meanwhile, MIC of Ala-Gly-Gly-Val-Pro-Arg against both bacterial species was 512 µg/mL. Our results suggest that peptides from sesame possess the ability to potentially hinder bacterial activity.

## 1. Introduction

Microorganisms are crucial elements in the assessment of food safety [1], and the issue of microbial food safety presents a significant obstacle for the expanding global food industry [2]. Traditional food preservation methods, food safety, and quality standards are not enough to meet the needs of consumers, while the excessive use of chemical preservatives, bacteria have developed drug resistance [3]. As a result, there is a pressing need to explore the potential of natural preservatives to maintain food quality. One promising solution is the use of antibacterial peptides (ABPs), which can safely and effectively preserve food without compromising its quality [4]. By harnessing the power of microorganisms and their antimicrobial products, we can develop biological preservatives that improve food safety while also prolonging shelf life [5].

ABPs are endogenous peptides that can be found in various organisms [6]. They are considered as an ancient molecule and are present in nearly all species [7]. ABPs typically comprise anywhere between 5 to 50 amino acids, and are characterized by their unique second order configurations such as α-helix or β-fold. These specific configurations lend functional specificity to each peptide [8,9]. ABPs are molecules that exhibit amphiphilic properties, as they possess both a hydrophilic end and a hydrophobic end [10]. Additionally, the majority of ABPs are cationic, which allows for electrostatic binding to specific anionic microbial targets [11]. It is noteworthy that the magnitude of positive charge varies among ABPs, and this net charge plays a crucial role in determining the affinity of binding to the targeted site of molecules and anions [12]. Research has demonstrated that ABPs acted by targeting the cytoplasmic membrane, inducing lipid bilayer structure depolarization, elevating membrane permeability, and ultimately leading to the eradication of microbes and cancer cells [13,14,15]. This method of action unfolds rapidly, making it challenging for pathogens to develop viable resistance [16].

Sulfonamides and sulfoxides are synthetic antimicrobial drugs that are well known for their efficacy [17]. Sulfonamides have been shown to selectively inhibit dihydropteroate synthase (DHPS), an essential enzyme in the folate pathway [18]. DHPS catalyzes the condensation reaction between p-aminobenzoic acid (pABA) and folic acid, leading to the formation of dihydropteroic acid (DHPt) via the condensation of pABA and 6-hydroxymethyl-7,8-dihydropterin pyrophosphate (DHPPP) [18,19]. Dihydropteroate synthase, being an essential enzyme for bacterial survival, can serve as a target for artificial screening of antimicrobial peptides [1]. The objective of our research was to discover small peptide inhibitors derived from sesquiterpenes that can effectively interact with the active site residues of DHPS [20]. Relatively speaking, the natural antimicrobial peptides reported in the literature can be broadly classified into two categories: (i) endogenous peptides already present in the plant matrix, and (ii) peptides produced from the plant proteome by treatment (e.g., enzymatic hydrolysis and fermentation) [21]. Various studies have long identified antimicrobial peptides from plant proteins, each of which required several steps to produce them [22]. This process involves selecting the appropriate enzyme and protein source, carrying out enzymatic hydrolysis, identifying bioactive peptides, validating their activity through in vivo and in vitro testing, and exploring their mechanism [23]. The rapid advancements in bioinformatics have resulted in computational techniques that utilize artificial intelligence prediction to detect bioactive peptides in proteins and guide the peptide synthesis process [24]. These techniques can accurately predict the tertiary structure of proteins and their interactions with ligands, as well as provide information on the stability of binding sites and protein–ligand complexes [25]. These methodologies not only provide cost effectiveness but also save valuable time and resources [26].

Sesame (*Sesamum indicum* L.) is a prominently cultivated oilseed crop that is renowned for its significant sesame protein levels ranging from 17% to 40% [27]. There have been reports indicating that sesame seed peptide has demonstrated significant efficacy in vitro and in vivo [28] in aspects such as antioxidant [29], blood lipid reduction [30], anti-aging, and diabetes resistance [27]. However, current documentation on the antibacterial activity of sesame peptides is limited [31].

The aim of our research was to identify potential peptide inhibitors derived from sesame protein that could interact with the active site residues of DHPS. In the current investigation, the identification and analysis of ABPs focused on two significant sesame protein through the utilization of bioinformatics tools. Furthermore, we implemented a virtual screening program based on docking. In addition, we have synthesized the selected antimicrobial peptides in vitro, tested their antimicrobial activity, and measured their minimum inhibitory concentration (MIC).

## 2. Materials and Methods

### 2.1. Analysis of the Sequence Similarity and Amino Acid Content of Sesame Sequences

Our study focused on the investigation of two primary proteins present in sesame, namely the 11S globulin seed storage protein 2 (Q9XHP0) and 2S seed storage protein 1 (Q9XHP1). We utilized UniProtKB (https://www.uniprot.org, accessed on 1 October 2020) to obtain relevant information such as the UniProtKB id, accession number, molecular mass, and amino acid composition of the sesame protein. Additionally, we utilized the ProtParam tool (http://web.expasy.org/protparam/, accessed on 2 October 2020) to assess the protein’s molecular weight, theoretical pI, aliphatic and GRAVY index, and amino acid composition.

### 2.2. In Silico Simulation of Sesame Protein Hydrolysis in the Gastrointestinal Tract

In the BIOPEP-UWM server (http://www.uwm.edu.pl/biochemia, accessed on 2 October 2020), the selection was made to target three prominent gastrointestinal proteases—pepsin (pH 1.3, EC 3.4.23.1), chymotrypsin A (EC 3.4.21.1) and trypsin (EC 3.4.21.4)—for the hydrolysis of sesame protein.

### 2.3. In Silico Prediction of the Antibacterial Activity of Released Peptides

Utilize CAMPR3 (http://www.camp.bicnirrh.res.in/, accessed on 4 October 2020) for selecting hydrolyzed peptides ranging from 2 to 15 amino acids and predict the produced peptides’ antibacterial efficacy. The application of multivariate statistical techniques such as random forest (RF), support vector machines (SVM), artificial neural network (ANN), and discriminant analysis (DA) was employed in CAMPR3. Notably, the ANN was utilized for the antibacterial activity.

### 2.4. In Silico Prediction of Physicochemical Properties of ABPs

The molecular mass, hydrophobic rate, and isoelectric point of sesame protein sequences were determined using the ToxinPred tool available at https://webs.iiitd.edu.in/raghava/toxinpred/index.html (accessed on 5 October 2020). Moreover, the net charge and water solubility of the peptide were predicted using the PepCalc tool, accessible at https://pepcalc.com/ (accessed on 5 October 2020).

### 2.5. Prediction of ABPs Toxicity and Allergenicity

We used ToxinPred (https://webs.iiitd.edu.in/raghava/toxinpred/index.html, accessed on 6 October 2020) and AllergenFP (http://ddg-pharmfac.net/AllergenFP/, accessed on 6 October 2020) as tools to determine the toxicity and sensitization of antimicrobial peptides, respectively. Threshold value (0.0) was used to differentiate between toxic and non-toxic peptides.

### 2.6. Molecular Docking

Based on the predicted results of antibacterial peptides, we decided to select cationic antibacterial peptide as ligands. As a positive control, we have selected the computer-designed Mycobacterium tuberculosis DHPS small peptide inhibitor Trp-Lys, which has been previously demonstrated as efficacious [32]. And the 3D crystal structure (PDB ID: 1AJ0) was obtained from Protein Data Bank (https://www.rcsb.org/, accessed on 7 October 2020). Docking simulation was used to better understand the interaction between the most effective ABPs with the binding sites of DHPS. In the docking experiment, we optimized the structure of small molecule ligands using the Avogadro software (Application Version: 1.2.0.) package, with the force fields set to MMFF94s for the minimization of ligand energy and subsequent merging of nonpolar hydrogen [33]. The receptor was prepared as follows: (1) Water molecules and ligands were removed from proteins; (2) Swiss-PdbViewer (Application Version: 4.10.) was used to check the protein structure for missing atoms, bonds, and contact points; (3) Autodock Tool (Application Version: 1.5.6.) was used to add polar hydrogen atom and Kollman charge in protein; (4) The getbox plug-in (https://github.com/MengwuXiao/Getbox-PyMOL-Plugin, accessed on 7 October 2020) was used to design the docking box (x = 41.5, y = 8.5, z = 2.0), covering all amino acid residues in the active part of the protein. AutoDock Vina (Application Version: 1.1.2.) was used to dock the ligands within the active sites, configured with 10 different conformations, the best ones were selected based on binding affinity (kcal/mol) and conformation. Finally, open-source PyMOL (Application Version: 2.4.0a0.) and Discovery Studio 2019 Client (Application Version: 19.1.0.) were utilized to combine conformation and intermolecular interactions and to discuss their interaction forces.

### 2.7. Strain Activation

The selected strains for in vitro validation consisted of Gram-negative *Escherichia coli* (*E. coli*) and Gram-positive *Staphylococcus aureus* (*S. aureus*), which were stored at −80 °C in glycerol. The strains were inoculated into the LB Nutrient Agar with the composition of peptone (10 g/L), sodium chloride (5 g/L), beef paste powder (3 g/L), and agar (15 g/L) using a plate scribing method in an ultra-clean bench, using an inoculation loop to pick up a loop of the strain. Following a 16 h incubation in a 37 °C environment, a ring of bacteria was inoculated into 100 mL of LB broth using an inoculation ring. Optimal conditions of 37 °C and 200 r/min were maintained in a shaker for 24 h to complete the experiment.

### 2.8. Plate Colony Count of Bacteria

In general, it is common for the concentration of the bacterial solution obtained after activation to be too high for direct quantification on a plate. As a result of past experience, it is typical to serially dilute the original bacterial solution in multiples of 10. Subsequently, 100 µL of the fifth, sixth, and seventh serial dilutions is incubated in a suitable Luria–Bertan nutrient agar medium. The plates are then left to incubate for 16 h before counting the colonies. Based on the obtained counting results, the concentration of the initial bacterial solution was ascertained to be 5 × 10^5^ colony-forming units per milliliter (CFU/mL).

### 2.9. MIC Measurement

The broth microdilution method was used in order to determine the MIC of the peptides (Ser-Thr-Ile-Arg, Ala-Gly-Gly-Val-Pro-Arg, and Trp-Lys) identified through virtual screening during the MIC testing process. The peptides were synthesized by Hangzhou ALL PEPTIDE Biology Co. During the experiment, we diluted the samples in gradient to six different concentrations ranging from 32 to 1024 µg/mL, with two parallel groups of replicates established for the sample group, a control group, and a blank group. Subsequently, each concentration of drug solution and an equal volume of bacterial solution were transferred to a 15 mL test tube, mixed thoroughly, and placed in a shaker set at 37 °C and 200 r/min for 24 h of incubation. After incubation, a 100 µL mixture was cultured on a Luria–Bertan nutrient agar medium for plate count. The MIC of the sample to the strain was determined by the absence of colony growth on the plate [34,35].

## 3. Results

### 3.1. Sesame Protein Sequence and Amino Acid Composition

Appendix A and Table 1 provide comprehensive information on sesame proteins, including their UniProtKB id, amino acid composition, molecular mass, theoretical pI, aliphatic, and GRAVY index predictions. Notably, the 11S globulin exhibits a theoretical maximum molecular weight of 51,840 Da, and its theoretical pI, aliphatic, and GRAVY index predictions are 7.73, 74.36, and −0.537, respectively. This protein contains 459 amino acids, with the most prevalent ones being Arg (R), Gln (Q), Ser (S), Leu (L), Ala (A), Glu (E), Gly (G), and Val (V). Meanwhile, the 2S globulin has a theoretical maximum molecular weight of 17,460 Da, and its theoretical pI, aliphatic, and GRAVY index predictions are 6.10, 45.54, and −0.589, respectively. This protein contains 148 amino acids, with the most prevalent ones being Glu (E), Gln (Q), Met (M), Arg (R), Ala (A), and Cys (C).

### 3.2. In Silico Gastrointestinal Enzymatic Hydrolysis of Sesame Proteins

To enhance the peptide yield obtained from these proteins, we opted to employ a multi-enzyme hydrolysis approach involving pepsin (pH 1.3, EC 3.4.23.1), chymotrypsin A (EC 3.4.21.1), and trypsin (EC 3.4.21.4), determined through BIOPEP-UWM online server. The outcomes of the simulated hydrolyzed proteins are illustrated in Figure 1. A total of 229 peptides were yielded from the two proteins, out of which 142 peptides were composed of 2 to 15 amino acids.

### 3.3. In Silico Prediction of Antimicrobial Activity and Physicochemical Properties of Peptides

The CAMPR3 database was utilized to predict the ABPs mentioned above, and the corresponding results are presented in Appendix A. Thirty-six distinct antimicrobial peptides were generated from sesame proteins, 11S globulin and 2S globulin. Multivariate statistical methods, such as random forest (RF), support vector machines (SVM), artificial neural network (ANN), and discriminant analysis (DA), were employed by CAMPR3 to predict these ABPs. Among various algorithms, ANN was used to classify the peptides as “ABP”. Based on the highest scoring algorithm value from CAMPR3 database, including SVM (1), RF (0.408), and DA (1), we selected and analyzed the ABPs sequence, as shown in Figure 2. Our study revealed that four peptides (EW, PL, EL, and PIF) had high confidence under all four algorithms, among which the SVM values were found to be “1”, RF values were “0.434, 0.478, 0.4935, 0.418”, and DA values were “0.99, 0.99, 0.791, 0.864”.

Appendix A summarizes the physicochemical properties of the ABPs in this study. Among the 36 peptides shown in our results, most of the peptides had a net charge between 0 and +1, and 6 of them were cationic antimicrobial peptides (Ser-Thr-Ile-Arg, Pro-Ser-Pro-Arg, Ser-Gln-Arg, Gly-Ser-Val-Arg, Ala-Gly-Gly-Val-Pro-Arg, and Val-Thr-Arg), and the theoretical isoelectric point (pI) of their peptides was 10.11, which was the highest value observed among the 36 peptides. Our analysis also showed that the hydrophobicity ratios of the 36 potential ABPs ranged from −0.9 to 0.67. The molecular weights of these peptides ranged from 374.47 to 555.71 Da.

### 3.4. Toxicity and Allergenicity Prediction of ABPs

The toxicity of the ingested peptide will affect the production of bioactive peptides, and if the ingested peptide (in its pure form) does not meet the demanding enzymatic conditions, then the beneficial effects of peptide catabolism on the host will be compromised. It becomes crucial to examine the toxicity and sensitization levels of antimicrobial peptides. In this paper, the hybrid model ToxinPred has been employed for toxicity prediction and results were shown in Appendix A. Antimicrobial peptides derived from sesame proteins are non-toxic and safe for food consumption, with a toxicity prediction value of 0. The online web server AllergenFP, which translates amino acid properties into fingerprint profiles, has been utilized to predict the allergenicity of amino acids. The application of this method to a set comprising 2427 known allergens and 2427 non-allergens has enabled the identification of 88% of the allergens, with a Mathews correlation coefficient of 0.759. The findings reveal that out of the 36 antimicrobial peptides discovered in sesame proteins, 28 are non-allergenic, and 8 are possible allergens. The latter group includes 7 from 11 s globulin and 1 from 2 s globulin.

### 3.5. Structure-Based Virtual Screening

Our objective is to target the DHPS and identify sesame-derived peptide inhibitors that are capable of interacting with the active site residues of DHPS. This will result in the inhibition of dihydrochloric acid synthase and ultimately lead to the death of bacterial cells.

In this study, we utilized molecular docking techniques to gain insights into the active site properties. Specifically, our investigation focused on the binding interactions between the compounds and amino acids within the active site of *E. coli* dihydropteroate synthase. For this purpose, the protein data bank file (PDB: 1AJ0) was chosen, and in this file, the enzyme dihydropteroate synthase is co-crystallized with sulfanilamide. The binding pocket is occupied by amino acids such as Arg 63, Arg 255, and Asn 22, forming a large network of hydrogen bonds. We selected six cationic antimicrobial peptides predicted from sesame proteins for virtual screening of DHPS. In Table 2, we present the docking binding affinities of seven cationic antimicrobial peptides, with values ranging from −6.4 to −8.1.

### 3.6. Molecular Interaction and Binding Mode

The results of our study involved an evaluation of the binding site interactions between the screened ligands and DHPS. The docking outcomes evinced the pivotal involvement of hydrogen bonding, hydrophobic forces, van der Waals forces, and attractive charge interplays in the intermolecular forces between sesame peptide and DHPS. As shown in Figure 3, the peptide Ser-Thr-Ile-Arg is bound in the hydrophobic pocket of DHPS via hydrogen bonding with PRO145, THR147, GLN149, LYS192, GLY189, SER222, SER219, and ARG235. Additionally, PHE190 exhibits hydrophobic interaction (Pi-Sigma), while PRO64 has electrostatic interaction (van der Waal) and heterotrimeric interaction (carbon hydrogen bond). Another polypeptide, Ala-Gly-Gly-Val-Pro-Arg, is bound in the hydrophobic pocket via hydrogen bonding with ARG255, PRO145, THR147, and GLY189. Hydrophobic interaction with ARG63, PRO64, PHE190 (Pi-Alkyl, Alkyl) and heterotrimeric interaction with GLN149 and LYS221 (carbon hydrogen bond) also contribute to the interaction forces.

### 3.7. Antibacterial Activity

The investigation analyzed the effects of three unique antimicrobial peptides on two diverse bacterial strains, *E. coli* and *S. aureus*. One of these peptides, short peptide Trp-Lys, is a potent inhibitor of *Mycobacterium tuberculosis* DHPS that has been reported using a computerized identification method, and its binding site overlaps with the binding region of pteridine monophosphate and sulfonamide drugs. Docking calculations also showed that this peptide has significantly higher potency than the sulfonamides, so we selected the short peptide Trp-Lys as a positive control for the experiment. Table 3 shows that Trp-Lys displayed favorable inhibition against *S. aureus* at an MIC of 256 μg/mL, but did not depict any notable inhibition against *E. coli*. Ser-Thr-Ile-Arg, on the other hand, demonstrated inhibitory effects on both bacterial strains, with MICs of 1024 μg/mL and 512 μg/mL, for *E. coli* and *S. aureus*, respectively. Ala-Gly-Gly-Val-Pro-Arg performed well in inhibiting both bacterial strains, with MICs of 512 μg/mL. According to the experimental outcomes, Ala-Gly-Gly-Val-Pro-Arg exhibited the most effective inhibition, with Ser-Thr-Ile-Arg following closely behind. Conversely, the short peptide Trp-Lys did not demonstrate any inhibition effect on Gram-negative *E. coli*, despite showing promising results on Gram-positive bacteria. These findings suggest that our virtual screening peptide has proven to be beneficial in effectively inhibiting bacterial growth.

## 4. Discussion

The objective of this research was to acquire bioactive peptides exhibiting antimicrobial characteristics from the principal proteins found in sesame seeds through a rapid and expedient approach. For this purpose, two antimicrobial peptides were identified using bioinformatics and various analytical tools. We synthesized two antimicrobial peptides and subsequently determined the minimum inhibitory concentrations of these peptides against Gram-positive *Staphylococcus aureus* and Gram-negative *Escherichia coli* by in vitro experiments. The results showed that the synthesized peptides (i.e., Ser-Thr-Ile-Arg and Ala-Gly-Gly-Val-Pro-Arg) inhibited the growth of both bacteria.

Sesame seeds contain two crucial proteins, namely 11S globulin (Alpha-globulin) and 2S globulin (Beta-globulin), which account for 80–90% of the total protein content [29]. In our investigation, we ascertained the precise amino acid sequences of these two proteins by utilizing the uniport database [36,37], 229 endogenous peptides were obtained by virtual enzyme digestion. We chose three proteases that are often present in the gastrointestinal tract for simultaneous enzyme digestion [38]. This decision was based on the evidence suggesting that hybrid enzyme hydrolysis yields more potent peptides when compared to the hydrolysis performed by a single enzyme [39]. According to reports, the reduction of amino acids has been found to enhance the discernment of antimicrobial peptides and their functional characteristics [40]. In light of this, we have opted to focus our investigation on peptides that consist of 2–15 amino acid residues, our examination will encompass the prediction of antimicrobial peptides, as well as the evaluation of their physicochemical properties, toxicity, and allergenicity [1]. Chemical synthesis is costly and laborious, and it is impossible to synthesize and validate all peptides one by one, which has certain limitations [41]. In contrast, bioinformatics offers a promising alternative by circumventing the intricate and time-consuming peptide purification process, thereby alleviating the burden associated with isolation and purification efforts [42].

Thirty-six peptides are predicted by bioinformatics to have possible antimicrobial activity. A similar analysis is previously performed, identifying to 11 peptides [31]. Further screening was conducted, taking into consideration the anticipated physicochemical properties, toxicity, and sensitization of the peptides. Toxicology may be one of the main reasons for the inability to produce biologically active peptides, and therefore, it is necessary to predict the toxicity of ABPs [43]. According to guidelines for evaluating the potential allergenicity of novel food proteins, a protein can be classified as a potential allergen if it possesses a minimum of 6–8 adjacent amino acid identities or exhibits a sequence similarity exceeding 35% when compared to established allergenic proteins within an 80 amino acid range [44]. In human and mammals, although ABPs also have anionic polypeptides, most ABPs are cationic polypeptides [45]. A positive charge is required for antimicrobial activity, and therefore, ABPs belong to the class of cationic peptides in the first place, and there is a close link between the cation activity of the peptide and the antimicrobial activity, as confirmed by many studies [46,47]. The greater the charge, the greater the antimicrobial activity against Gram-negative and Gram-positive pathogens [48]. Therefore, in our study, we selected non-toxic, non-sensitizing, and positively charged peptides for molecular docking screening. According to recent research, a powerful inhibitor (short peptide Trp-Lys) of *Mycobacterium tuberculosis* DHPS has been discovered [32]. This inhibitor is capable of binding to both pteridine monophosphate and sulfa drug-binding regions [17]. As the inhibitor attaches to a remarkably persistent pterin binding site, it has a likelihood of encountering fewer drug resistance mutations [32]. Therefore, we chose Trp-Lys as a positive control, based on the binding affinity values analysis, it was observed that Ala-Gly-Gly-Val-Pro-Arg (−8.1 kcal/mol) and Ser-Thr-Ile-Arg (−8.0 kcal/mol) peptides outperformed the positive control (Trp-Lys), indicating their potential as viable ABPs [38,49].

Hence, to validate the peptides acquired, we conducted inhibition experiments and determined the minimum inhibitory concentration of said peptides, and our primary objective was to identify a bacteriostatic agent [17]. The inhibitory potential of our peptide against the Gram-positive bacterium *Staphylococcus aureus* and the Gram-negative bacterium *Escherichia coli* has been confirmed through experimental validation. As mentioned earlier, the antimicrobial activity of peptides can be attributed to the net positive charge in their structure [45]. They allow the peptide to react with the negatively charged membrane phospholipids of bacteria and insert into the cell membrane, which leads to increased membrane permeability of the bacteria and cell death, and cell morphology is altered [50]. In fact, electrostatic attraction between ABPs and negatively charged microbial membranes due to the net positive charge, while hindering interaction with neutrally charged mammalian cell membranes, and many studies have shown a correlation between the charge of ABPs and antimicrobial activity [51]. And we also think the strong positive charge leads to a high affinity for interaction with bacterial membranes, and improved antibacterial action to some extent [52]. The positive charge of Ser-Thr-Ile-Arg and Ala-Gly-Gly-Val-Pro-Arg peptides can be attributed to the presence of the Arg amino acid within these sequences. It is documented that increasing the number of arginine residues in peptide sequences can enhance antibacterial activity, and this enhancement is due to the positive charge interaction between arginine residues and phospholipids, leading to increased membrane permeabilization [53]. Liu [54] and colleagues reported a peptide sequence, Arg-Ser-Ser, which showed that Arg-Ser-Ser treatment increased zeta potential, hydrophobicity, and cell membrane permeability at the cell surface, suggesting that the positively charged Arg-Ser-Ser may interact with negatively charged bacterial membranes. The hydrophobicity of ABPs is another distinguishing feature, and the lipid bilayer that makes up the microbial membrane is hydrophobic in nature [55]. The hydrophobic residues also promote the interaction of the peptide with phospholipid membranes, increasing the bactericidal effect of the peptide [56] Ser-Thr-Ile-Arg and Ala-Gly-Gly-Val-Pro-Arg due to the presence of hydrophobic amino acid residues Ile, Ala, and Pro, whose presence contributes to the efficient inhibitory activity of this peptide [57]. This may also explain the stronger bacterial inhibition of the peptide Ala-Gly-Gly-Val-Pro-Arg than Ser-Thr-Ile-Arg. Emelianova et al. [58] demonstrated that the ChMAP-28 peptide derived from goat protein induced cancer cell necrosis through plasma membrane disruption, they also observed that the peptide permeabilized the cytoplasmic membrane of HL-60 human leukemia cells. Zhou [59], on the other hand, revealed that lugensins, another peptide, hindered bacterial growth by disrupting bacterial membranes. Further exploration and verification are required to ascertain whether the peptides identified in this study also adhere to the same mechanism of action.

## 5. Conclusions

The use of computational screening facilitates the rapid and proficient detection of potential peptides with antimicrobial properties that can then be experimentally evaluated, thereby speeding up the biopreservative discovery process. In this study, we screened and synthesized two stable, non-allergenic, non-toxic peptides (Ser-Thr-Ile-Arg and Ala-Gly-Gly-Val-Pro-Arg) from gastrointestinal hydrolysates of sesame seeds, which have great potential for research and development. They were verified as antibacterial peptides by using in vitro experiments. The docking analysis revealed that hydrogen bonding, hydrophobic interaction, van der Waals force, and attractive charge interaction force are significant factors contributing to the interaction force between sesame peptides and DHPS. Both *E. coli* and *S. aureus* were effectively suppressed through the utilization of either Ser-Thr-Ile-Arg or Ala-Gly-Gly-Val-Pro-Arg. Specifically, the MIC values of Ser-Thr-Ile-Arg for *E. coli* and *S. aureus* were 1024 µg/mL and 512 µg/mL, respectively. Meanwhile, Ala-Gly-Gly-Val-Pro-Arg exhibited MIC of 512 µg/mL against both these bacterial species. In summary, the two peptides demonstrated a moderate level of antibacterial potency. Peptides derived from sesame protein obtained by gastrointestinal protease hydrolysis have the potential to exhibit antibacterial activity, although in vivo experiments are needed for subsequent verification.

## Figures and Tables

**Figure 1 nutrients-16-00175-f001:**
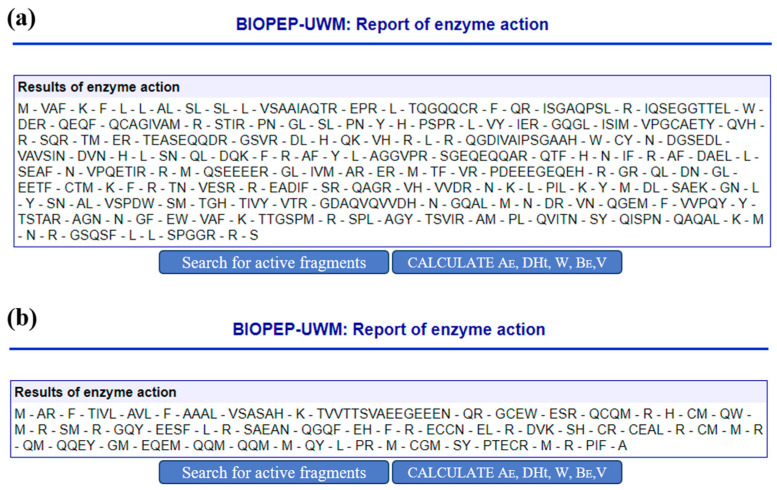
Peptides from (**a**) 11S globulin; (**b**) 2S globulin by gastrointestinal digestion simulated by pepsin, chymotrypsin A, and trypsin in the BIOPEP database.

**Figure 2 nutrients-16-00175-f002:**
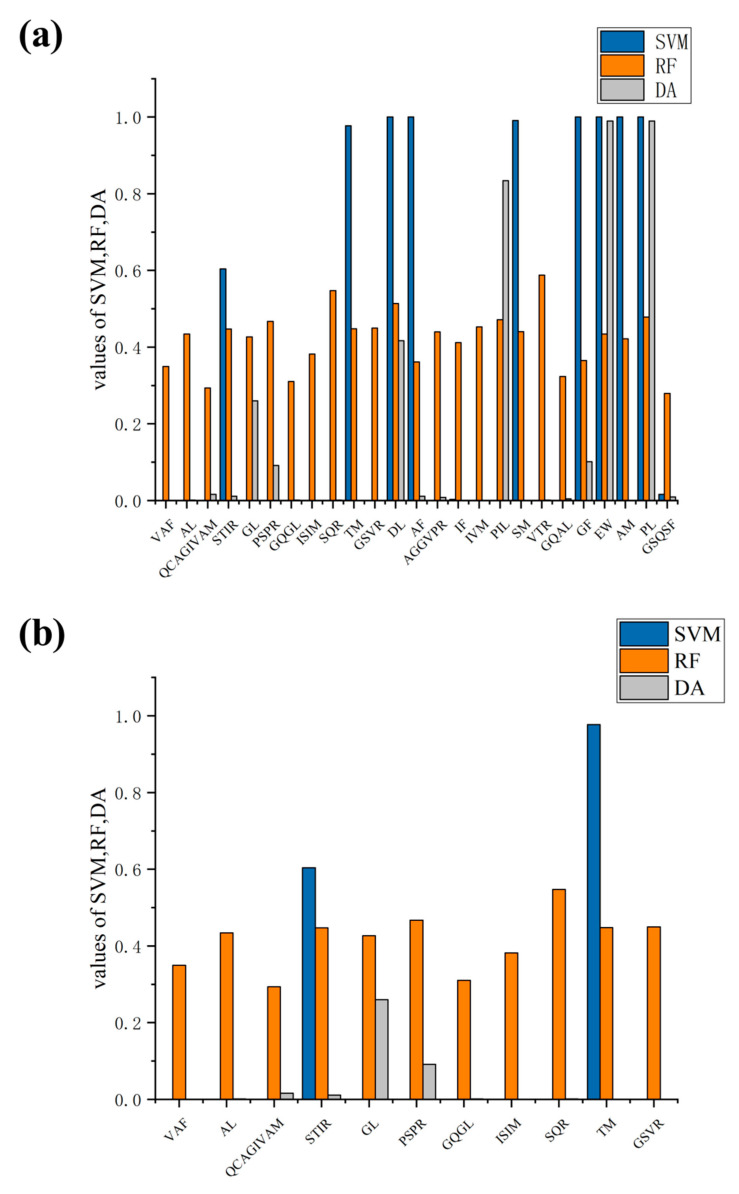
Prediction of antimicrobial peptides by statistical models in CAMPR3 database including random Forest (RF), support vector machines (SVM), artificial neural network (ANN), and discriminant analysis (DA); (**a**) 11S globulin, (**b**) 2S globulin.

**Figure 3 nutrients-16-00175-f003:**
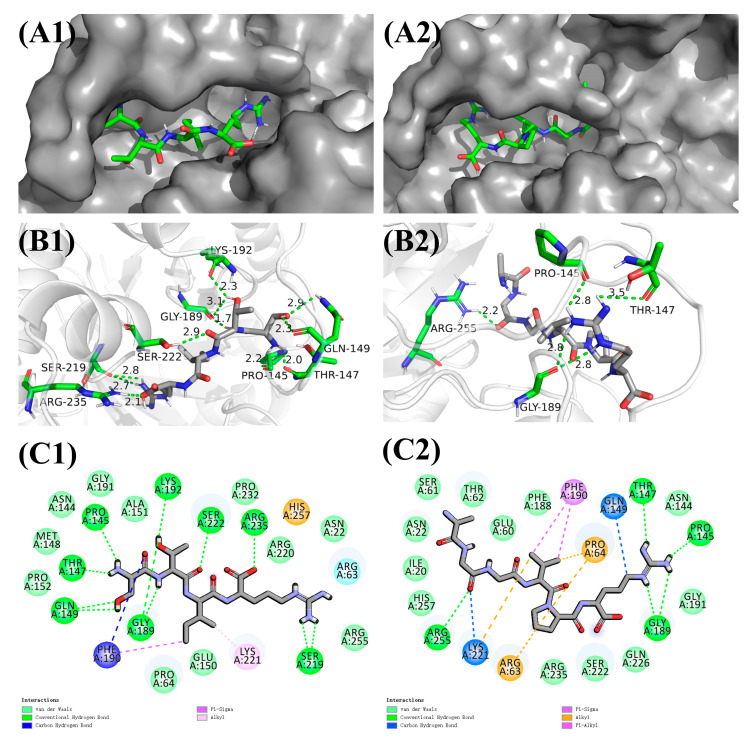
Molecular docking analysis: best-chosen conformation of the binding sites of (**A1**) Ser-Thr-Ile-Arg and (**A2**) Ala-Gly-Gly-Val-Pro-Arg in the docking pocket, where the ligand is represented by a green stick and the protein by a gray surface; (**B1**) 3D view of Ser-Thr-Ile-Arg and (**B2**) Ala-Gly-Gly-Val-Pro-Arg peptide binding site interactions, where the ligand is indicated in gray and the hydrogen bonding with the protein amino acid is indicated in green; (**C1**) 2D view of the interaction of Ser-Thr-Ile-Arg and (**C2**) Ala-Gly-Gly-Val-Pro-Arg binding sites.

**Table 1 nutrients-16-00175-t001:** Amino acid composition, molecular mass, theoretical pI, aliphatic, and GRAVY index predictions of sesame proteins.

Amino Acid Composition	11S Globulin Seed Storage Protein 2	2S Seed Storage Protein 1
Ala (A)	34	12
Arg (R)	43	15
Asn (N)	21	3
Asp (D)	18	1
Cys (C)	5	10
Gln (Q)	38	16
Glu (E)	33	18
Gly (G)	33	6
His (H)	11	4
Ile (I)	20	2
Leu (L)	35	7
Lys (K)	9	2
Met (M)	15	16
Phe (F)	18	6
Pro (P)	19	3
Ser (S)	36	9
Thr (T)	23	5
Trp (W)	4	2
Tyr (Y)	12	4
Val (V)	32	7
Number of amino acids	459	148
Theoretical pI	7.73	6.10
Aliphatic index	74.36	45.54
GRAVY index	−0.537	−0.589

**Table 2 nutrients-16-00175-t002:** Docking binding affinity of six cationic antimicrobial peptides.

Peptides	Binding Energy (kcal/mol)	The Interacting Residues of Binding Site and the Specific Interaction for Each Ligand Considered
Ser-Thr-Ile-Arg	−8.0	ASN22, PRO64, ASN144, MET148, GLU150, ALA151, PRO152, GLY191, ARG220, PRO232, ARG255 (van der waals); PRO145, THR147, GLN149, GLY189, LYS192, SER219, SER222, ARG235 (Conventional Hydrogen Bond); PHE190 (Carbon Hydrogen Bond); PHE190 (Pi-Sigma); LYS221 (Alkyl)
Pro-Ser-Pro-Arg	−7.1	THR62, ARG63, PRO64, MET141, PHE188, PHE190, LYS192, ARG220, GLN226, ARG255 (van der waals); GLN142, ASN144, PRO145, THR147, GLY189, SER222 (Conventional Hydrogen Bond); LYS221, SER222 (Carbon Hydrogen Bond)
Ser-Gln-Arg	−7.2	ARG63, PRO64, ASN144, GLU150, ALA151, PHE190, ASN193, ASN197, MET223 (van der waals); THR62, PRO145, THR147, GLN149, GLY189, GLY191, LYS192, SER222, GLN226 (Conventional Hydrogen Bond); GLN142 (Carbon Hydrogen Bond)
Gly-Ser-Val-Arg	−7.2	ARG63, GLY143, MET148, GLU150, ALA151, PRO152, PHE190, GLY191, ASN193, ASN197, GLN226 (van der waals); THR62, ASN144, PRO145, GLN149, GLY189, LYS192, SER222 (Conventional Hydrogen Bond); GLN142, THR147 (Carbon Hydrogen Bond); PRO64 (Alkyl)
Ala-Gly-Gly-Val-Pro-Arg	−8.1	ILE20, ASN22, GLU60, SER61, THR62, ASN144, PHE188, GLY191, SER222, GLN226, ARG235, HIS257 (van der waals); PRO145, THR147, GLY189, ARG255 (Conventional Hydrogen Bond); GLN149, LYS221 (Carbon Hydrogen Bond); PHE190 (Pi-Sigma); ARG63, PRO64 (Alkyl); PHE190 (Pi-Alkyl)
Val-Thr-Arg	−6.4	MET141, GLY143, ASN144, GLN149, GLU150, ALA151, PRO152, PHE190, GLY191, ASN193, GLN226 (van der waals); GLN142, THR147, GLY189, LYS192 (Conventional Hydrogen Bond); PRO145, SER222 (Carbon Hydrogen Bond); PRO64, MET148 (Alkyl)
Trp-Lys	−7.2	PRO64, GLY143, ASN144, PRO145, GLN149, ALA151, PRO152, PHE190, SER222, ARG235 (van der waals); GLN142, GLY189, ARG220 (Conventional Hydrogen Bond); ARG63, LYS221 (Pi-Alkyl); LYS221 (Amide-Pi Stacked)

**Table 3 nutrients-16-00175-t003:** Minimal inhibitory concentration (MIC) values of Trp-Lys, Ser-Thr-Ile-Arg, and Ala-Gly-Gly-Val-Pro-Arg.

Bacteria	MIC Values of Trp-Lys (μg/mL)	MIC Values of Ser-Thr-Ile-Arg (μg/mL)	MIC Values of Ala-Gly-Gly-Val-Pro-Arg (μg/mL)
*E. coli*	~	1024	512
*S. aureus*	256	512	512

~, no antibacterial ability.

## Data Availability

Data are contained within the article and Appendix A.

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
