# Peer review of "Antibacterial Effect of Sesame Protein-Derived Peptides against *Escherichia coli* and *Staphylococcus aureus*: In Silico and In Vitro Analysis"

_nutrients, 2024, doi:10.3390/nu16010175_

Round 1
Reviewer 1 Report
Comments and Suggestions for Authors
The manuscript with ID nutrients-2733192, titled Antibacterial effect of sesame protein derived peptides against Escherichia coli and Staphylococcus aureus: in silico and in vitro analysis’ reports the potential use of peptides from sesame as antibacterial in food without compromise the quality. The approach employed, predicts both in silico and in vitro experiments. All the in silico evaluations carried out in this study confer originality to the manuscript and suggest the opportunity to avoid a lot of lab experiments (often useless) obtaining punctual predictions that can be used in the in vitro experiments.
Only in the table 2, I kindly suggest to add a column with the interacting residues of binding site and the specific interaction for each ligands considered.
Author Response
Response to Reviewers
Dear Editor and Reviewers,
Thanks for your great concern and comments on my initial manuscript. Complying with your comments, we have revised our manuscript carefully. The revised parts are marked in red for your convenience of re-reviewing. The answers to each comment have been addressed as follows.
Reviewer #1:
Q1: Only in the table 2, I kindly suggest to add a column with the interacting residues of binding site and the specific interaction for each ligands considered.
Answer: Thanks very much for your comments. According to your suggestion, we have added a column in Table 2 indicating the interacting residues of binding site and the specific interaction for each ligands considered. See lines 259-260.

Reviewer 2 Report
Comments and Suggestions for Authors
The topic of the article results very interesting and elaborate. Using computer tools to predict experimental effects is a good way to focus the research aim in less time and with less cost.
Why wasn't the minimal bactericidal concentration also measured? It would also have been interesting to see if there was such an effect given by the synthesized antimicrobial peptides.
The discussion section would seem not to be there because it does not appear in any headline. In the results, however, parts of the disccussion appear in some chapters. Perhaps the authors intended to do results and discussion together? If so, however, each results should be discussed so the authors should change the title to this section and implement the missing parts.
Minor comments
-Title, abstract, keywords, article: All scientific names of bacteria and plants (and their abbreviations )should be written in italics, please correct them (Escherichia coli, Staphylococcus aureus, E. coli, S. aureus, Sesamum indicum, Mycobacterium tuberculisis, Pseudomonas aeruginosa, P. aeruginosa)
- Title, abstract, article, supplementary material: “in vitro”, “in vivo” and “in silico” should be written in italics, please correct them. Where the title is written in italics such words should not be written in italics.
- Pages 1-2, 8-9 lines 40-41, 41-43, 43-44, 44-45, 45-47, 49-52, 54-55, 55-56, 68-71, 71-73, 73-75, 75-76, 81-82, 225-227, 240-243, 243-245, 245-248, 253-254, 302-303, 303-305, Many bibliographical references are missing. Every statement in the introduction and discussion sections must be supported by the literature. Please add them.
- Page 2 lines 56-61 this sentence is too long, please rewrite it.
- Page 3 line 133 please add Avogadro software references.
- Page 4 line 148 What medium and with what composition was used for the bacteria to resume?
-Page 4 line 151 the liquid medium is this Luria Bertan medium? Why not write it down directly? So it's a little confusing.
-Page 4 MIC measurement section: Why wasn't the minimum bactericidal concentration measured? If it is possible I suggest doing it. It is a small step after MIC that can add value to the work.
-Page 4 lines 182-183 and 186-187 the ammino acid names were written in different ways please use the same way to avoid confusion in the reader.
-Page 5 lines 192-193 this part belongs to the discussion section. You don't put bibliographic references in the results.
-Figure 2 are there any statistical differences? if there are share them on the histogram bars.
-Pages 7-8 lines 222-259 this part belongs to the discussion section. You don't put bibliographic references in the results.
- Page 8 lines 274-276 this part belongs to the conclusion section. Please move it.
- Page 9 lines 288-289, 295-297 these parts belong to the discussion section.
-Page 9 lines 291-293 this part belongs to the materials and methods section. Please move it.
-Figure 3 The pictures are too small, you can't read the lettering and captions inside. Please enlarge the image or arrange it vertically.
-Pages 10-11 lines 339-363 this part belongs to the discussion section.
-Page 10 line 343 “Pseudomonas aeruginosa” the first time a bacterium is mentioned the name should be spelled out in full. please correct.
Comments on the Quality of English Language
I suggest reviewing the English language.
Author Response
Response to Reviewers
Dear Editor and Reviewers,
Thanks for your great concern and comments on my initial manuscript. Complying with your comments, we have revised our manuscript carefully. The revised parts are marked in red for your convenience of re-reviewing. The answers to each comment have been addressed as follows.
Reviewer #2:
- Why wasn't the minimal bactericidal concentration also measured? It would also have been interesting to see if there was such an effect given by the synthesized antimicrobial peptides.
Answer: Thanks very much for your comments. During the preparation of the preliminary experiments, we read the articles about the antibacterial activity of peptides [1–3], and they all carried out the determination of the minimum inhibitory concentration, which could verify the antimicrobial activity of the peptides. And in the molecular docking experiment, we benchmarked sulfonamides, which mainly inhibit the mode of action of bacteria through competitive inhibition of bacterial DNA synthesis [4], it can only inhibit the growth of bacteria but can not completely kill bacteria. Therefore, in the course of the subsequent experiments, we considered only the minimum inhibitory concentration of the experiment. Thank you again for your suggestion, we will improve it in subsequent experiments.
Reference:
- Okella, H.; Okello, E.; Mtewa, A.G.; Ikiriza, H.; Kaggwa, B.; Aber, J.; Ndekezi, C.; Nkamwesiga, J.; Ajayi, C.O.; Mugeni, I.M.; et al. ADMET Profiling and Molecular Docking of Potential Antimicrobial Peptides Previously Isolated from African Catfish, Clarias Gariepinus. Front. Mol. Biosci.2022, 9, 1039286, doi:10.3389/fmolb.2022.1039286.
- Hu, Y.; Ling, Y.; Qin, Z.; Huang, J.; Jian, L.; Ren, D.F. Isolation, Identification, and Synergistic Mechanism of a Novel Antimicrobial Peptide and Phenolic Compound from Fermented Walnut Meal and Their Application in Rosa Roxbughii Tratt Spoilage Fungus. Food Chemistry2024, 433, 137333, doi:10.1016/j.foodchem.2023.137333.
- Xu, Y.; Guo, W.; Luo, D.; Li, P.; Xiang, J.; Chen, J.; Xia, X.; Xie, Q. Antibiofilm Effects of Punicalagin against Staphylococcus Aureus in Vitro. Front. Microbiol.2023, 14, 1175912, doi:10.3389/fmicb.2023.1175912.
- Ovung, A.; Bhattacharyya, J. Sulfonamide Drugs: Structure, Antibacterial Property, Toxicity, and Biophysical Interactions. Biophys Rev2021, 13, 259–272, doi:10.1007/s12551-021-00795-9.
- The discussion section would seem not to be there because it does not appear in any headline. In the results, however, parts of the disccussion appear in some chapters. Perhaps the authors intended to do results and discussion together? If so, however, each results should be discussed so the authors should change the title to this section and implement the missing parts.
Answer: Thanks very much for your comments, which are very helpful for us to improve the manuscript. We have revised the whole manuscript carefully and added discussion section. See page 11-13 lines 291-375.
- Title, abstract, keywords, article: All scientific names of bacteria and plants (and their abbreviations )should be written in italics, please correct them (Escherichia coli, Staphylococcus aureus, E. coli, S. aureus, Sesamum indicum, Mycobacterium tuberculisis, Pseudomonas aeruginosa, P. aeruginosa)
Answer: Thank you very much for your comment. Regarding your reference to the fact that all scientific names of bacteria and plants (and their abbreviations) should be written in italics, we have made a change in the article. See page 1 line 3, lines 13-14, lines 20-21, line 25; page line 79; page 3 line 125; page 4 lines 143-144; page 11 lines 273-274, lines 279-282, line 285, lines 308-309; page 12 lines 358-359; page 13 lines 385-387.
- Title, abstract, article, supplementary material: “in vitro”, “in vivo” and “in silico” should be written in italics, please correct them. Where the title is written in italics such words should not be written in italics.
Answer: Thank you very much for your comment. Regarding your mention that all "in vitro", "in vivo" and "in silico" should be italicized, we have made the change in the article. See page 1 line 3, line 14, line 20; page 2 line 80, line 88; page 4 line 143; page 11 lines 288-289, lines 297-298; page 13 line 382, line 391, line 395.
- Page 2 lines 56-61 this sentence is too long, please rewrite it.
Answer: Thanks very much for your comments. Based on your suggestion, we have rewritten it. See page 2 lines 56-6..
DHPS catalyzes the condensation reaction between p-aminobenzoic acid (pABA) and folic acid, leading to the formation of dihydropteroic acid (DHPt) via the condensation of pABA and 6-hydroxymethyl-7,8-dihydropterin pyrophosphate (DHPPP) [18,19].Dihydropteroate synthase, being an essential enzyme for bacterial survival, can serve as a target for artificial screening of antimicrobial peptides [1].
- Page 3 line 133 please add Avogadro software references.
Answer: Thanks very much for your comments. Following your suggestion, we have added the Avogadro software reference. See page 3 line 131.
- Page 4 line 148 What medium and with what composition was used for the bacteria to resume?
Answer: Thank you very much for your comment. We used LB Broth for strain recovery, consisting of peptone (10 g/L), sodium chloride (5 g/L), beef paste powder (3 g/L) and agar (15 g/L), with modifications in the manuscript See page 4 lines 145-146.
- Page 4 line 151 the liquid medium is this Luria Bertan medium? Why not write it down directly? So it's a little confusing.
Answer: Thank you very much for your comment. As per your question, the liquid medium is LB broth and we have modified the manuscript as per your suggestion. See page 4 line 149.
- Page 4 MIC measurement section: Why wasn't the minimum bactericidal concentration measured? If it is possible I suggest doing it. It is a small step after MIC that can add value to the work.
Answer: Thank you very much for your comment. In the molecular docking experiment, we benchmarked sulfonamides, which mainly inhibit the mode of action of bacteria through competitive inhibition of bacterial DNA synthesis [1], it can only inhibit the growth of bacteria but can not completely kill bacteria. Therefore, in the course of the subsequent experiments, we considered only the minimum inhibitory concentration of the experiment. Thank you again for your suggestion, we will improve it in subsequent experiments.
Reference:
- Ovung, A.; Bhattacharyya, J. Sulfonamide Drugs: Structure, Antibacterial Property, Toxicity, and Biophysical Interactions. Biophys Rev 2021, 13, 259–272, doi:10.1007/s12551-021-00795-9.
- Page 4 lines 182-183 and 186-187 the ammino acid names were written in different ways please use the same way to avoid confusion in the reader.
Answer: Thank you very much for your comment. Based on your suggestion, we have harmonized the manuscript expressions related to peptides throughout the manuscript. See page lines 19-23; page 4 line 164; page 6 lines 213-215; page 9 lines 254-255; page 10 line 261, line 265, line 272; page 17 lines 285-288, lines 292-294, line 304; page 12 line 338, lines 342-344, line 360, line 367, line 372; page 13 lines 374-375, line 386-387, lines 393-395.
- Page 5 lines 192-193 this part belongs to the discussion section. You don't put bibliographic references in the results.
Answer: Thank you very much for your comment. Based on your suggestion, we have moved the section to the Discussion section and removed the references in the Results section.
To enhance the peptide yield obtained from these proteins, we opted to employ a multi-enzyme hydrolysis approach involving pepsin (pH 1.3, EC 3.4.23.1), chymotrypsin A (EC 3.4.21.1), and trypsin (EC 3.4.21.4), determined through BIOPEP-UWM online server. The outcomes of the simulated hydrolyzed proteins are illustrated in Figure 1. A total of 229 peptides were yielded from the two proteins, out of which 142 peptides were composed of 2 to 15 amino acids.
- Figure 2 are there any statistical differences? if there are share them on the histogram bars.
Answer: Thank you very much for your comment. Figure 2 shows the antimicrobial peptides predicted by the statistical model in the CAMPR3 database, with the same score for each prediction, and therefore not statistically different.
- Pages 7-8 lines 222-259 this part belongs to the discussion section. You don't put bibliographic references in the results.
Answer: Thank you very much for your comment. Based on your suggestion, we have moved this section to the Discussion section. And we have modified the results section by removing the references in the results.
Table S3 summarizes the physicochemical properties of the ABPs in this study. Among the 36 peptides shown in our results, most of the peptides had a net charge between 0 and +1, and six of them were cationic antimicrobial peptides (STIR, PSPR, SQR, GSVR, AGGVPR, and VTR), and the theoretical isoelectric point (pI) of their peptides was 10.11, which was the highest value observed among the 36 peptides. peptides, which is the highest value observed among the 36 peptides. Our analysis also showed that the hydrophobicity ratios of the 36 potential ABPs ranged from -0.9 to 0.67. The molecular weights of these peptides ranged from 374.47 to 555.71 Da.
- Page 8 lines 274-276 this part belongs to the conclusion section. Please move it.
Answer: Thank you very much for your comment. Based on your suggestion, we have moved this part to the conclusion section and And we have also removed the references in the results. See page 8 line 238.
- Page 9 lines 288-289, 295-297 these parts belong to the discussion section.
Answer: Thank you very much for your comment. Based on your suggestion, we have moved this part to the discussion section. See page 12 line 331, line 345.
- Page 9 lines 291-293 this part belongs to the materials and methods section. Please move it.
Answer: Thank you very much for your comment. Based on your suggestion, we have moved this part to the materials and methods section. See page 3 line 139.
- Figure 3 The pictures are too small, you can't read the lettering and captions inside. Please enlarge the image or arrange it vertically.
Answer: Thank you very much for your comment. Based on your suggestion, we have arranged the images vertically so that readers can see the content of the images. See page 10 line 270.
- Pages 10-11 lines 339-363 this part belongs to the discussion section.
Answer: Thank you very much for your comment. Based on your suggestion, we have moved this section to the Discussion section. And we have also removed the references in the results. See page 11 line 275.
- Page 10 line 343 “Pseudomonas aeruginosa” the first time a bacterium is mentioned the name should be spelled out in full. please correct.
Answer: Thank you very much for your comment. However, in the revised manuscript, we have removed the reference in this article, and deleted the content of “Pseudomonas aeruginosa” as well.
- Comments on the Quality of English Language. I suggest reviewing the English language.
Answer: Thanks very much you for your comments, which are very helpful for us to improve the manuscript. We have revised the whole manuscript carefully and tried to avoid any grammar or syntax error. In addition, we have asked several friends who live in USA and are skilled authors of English-language paper to check the English. We hope that the revised paper will be more clear and accurate on expressions.

Reviewer 3 Report
Comments and Suggestions for Authors
This manuscript investigates the antimicrobial potential of peptides derived from sesame protein using both in silico and in vitro approaches. It presents intriguing results, although the concentration of the work appears somewhat elevated. Below are some recommendations to improve the work
#1 Comments to the Authors
The introduction is comprehensive but could benefit from a more focused discussion in the results section. It tends to be lengthy and occasionally verbose. A recommendation is made to revisit this section, considering the reiterated objective and information that could be better placed earlier. Additionally, the restatement of the objective at the end of this section should be addressed.
#2 Comments to the Authors
Lines 86-90: Suggest transferring this content to the methods section.
#3 Comments to the Authors
Line 161: Specify the number of colonies selected for the assay.
#4 Comments to the Authors
Section 2.9 MIC Measurement: Clarify the reference used for the test method, as it is currently unclear.
#5 Comments to the Authors
Line 230: In the results/discussion, consider addressing the difficulty microorganisms face in developing resistance to antimicrobial peptides that interact with the membrane. Evolutionarily, drastic changes in membrane composition incur high energetic costs and risks to cell viability. Some relevant studies discuss this aspect, and it would be valuable to highlight it, as this constitutes the primary known mechanism of action of antimicrobial peptides.
#6 Comments to the Authors
Section 3.4 Toxicity and Allergenicity Prediction of ABPs: Draw a parallel between the potential toxicity of certain antibiotics, like ciprofloxacin, and the results found in this session. This could pave the way for possible synergism experiments.
#7 Comments to the Authors
Section 3.7 Antibacterial Activity: Discuss the possible mechanisms of action of these peptides in more depth. Consider additional experiments to analyze morphological changes in cells. The current discussion of results is deemed superficial.
Author Response
Response to Reviewers
Dear Editor and Reviewers,
Thanks for your great concern and comments on my initial manuscript. Complying with your comments, we have revised our manuscript carefully. The revised parts are marked in red for your convenience of re-reviewing. The answers to each comment have been addressed as follows.
Reviewer #3:
- The introduction is comprehensive but could benefit from a more focused discussion in the results section. It tends to be lengthy and occasionally verbose. A recommendation is made to revisit this section, considering the reiterated objective and information that could be better placed earlier. Additionally, the restatement of the objective at the end of this section should be addressed.
Answer: Thanks very much for your comments and compliments on the introduction section. We have reconsidered and streamlined the results section, adding the highlights to the discussion section.
- Lines 86-90: Suggest transferring this content to the methods section.
Answer: Thank you very much for your comment. Based on your suggestion, we have moved the section to the methods section and made changes to this section. See page 2 line 85.
- Line 161: Specify the number of colonies selected for the assay.
Answer: Thank you very much for your comment. In our experiments, we chose a colony number of 5 × 105 colony-forming units per milliliter (CFU/mL). We have added the missing information in page 4 line 160.
- Section 2.9 MIC Measurement: Clarify the reference used for the test method, as it is currently unclear.
Answer: Thank you very much for your comment. Based on your suggestion, we have clarified the references used for the MIC detection method. See page 4 line 173.
- Line 230: In the results/discussion, consider addressing the difficulty microorganisms face in developing resistance to antimicrobial peptides that interact with the membrane. Evolutionarily, drastic changes in membrane composition incur high energetic costs and risks to cell viability. Some relevant studies discuss this aspect, and it would be valuable to highlight it, as this constitutes the primary known mechanism of action of antimicrobial peptides.
Answer: Thank you very much for your comment. Based on your suggestion, we discussed the structural properties of the peptides in comparison with the existing literature, which led us to indirectly hypothesize that the mechanism of antimicrobial action of the peptides found in this study may act on the cell membranes of microorganisms, which we consider to be a significant finding. We will further explore the relationship between bacteria and cell membranes in future experimental studies. See page 13-14, lines 350-381.
- Section 3.4 Toxicity and Allergenicity Prediction of ABPs: Draw a parallel between the potential toxicity of certain antibiotics, like ciprofloxacin, and the results found in this session. This could pave the way for possible synergism experiments.
Answer: Thank you very much for your comment. Based on your suggestion, we offer the following answers: ToxinPred is a unique computerized method that can be used to predict the toxicity of peptides/proteins [1]. The current calculations cannot predict the toxicity of antibiotics, and we will further investigate the synergistic relationship between antibiotics and peptides in future studies.
References:
- Gupta, S.; Kapoor, P.; Chaudhary, K.; Gautam, A.; Kumar, R.; Open Source Drug Discovery Consortium; Raghava, G.P.S. In Silico Approach for Predicting Toxicity of Peptides and Proteins. PLoS ONE 2013, 8, e73957, doi: 10.1371/journal.pone.0073957.
- Section 3.7 Antibacterial Activity: Discuss the possible mechanisms of action of these peptides in more depth. Consider additional experiments to analyze morphological changes in cells. The current discussion of results is deemed superficial.
Answer: Thank you very much for your comment. Based on your suggestion, we discuss the possible mechanisms of action of these peptides in more depth in the Discussion section. In our next studies, we will consider the relationship between the peptide and the cell membrane, with implications for changes in bacterial morphology. See page 13-14, lines 350-381.

Round 2
Reviewer 2 Report
Comments and Suggestions for Authors
I appreciated the great work the authors did on the manuscript, and I think it is improved by it.
I have some minor comments:
- Figure 3: Into the figure, the last line of image it is difficult to read what the different colors, in which the amino acids are contained, indicate. I recommend dividing for lines the figure in a, b ,c and, indicating in the caption what the different colors mean.
- line 327 What does it mean “realm of allergenicity”? Please correct it.
- line 333 What does it mean “cation city”? Please correct it.
- line 396 I suggest “both these bacterial species” instead of “two bacterial species”.
Comments on the Quality of English LanguageMinor editing of English language required
Author Response
Response to Reviewers
Dear Editor and Reviewers,
Thanks for your great concern and comments on my initial manuscript. Complying with your comments, we have revised our manuscript carefully. The revised parts are marked in red for your convenience of re-reviewing. The answers to each comment have been addressed as follows.
Reviewer #2:
- Figure 3: Into the figure, the last line of image it is difficult to read what the different colors, in which the amino acids are contained, indicate. I recommend dividing for lines the figure in a, b ,c and, indicating in the caption what the different colors mean.
Answer: Thanks very much for your comments. Based on your suggestion, we have changed Figure 3 and added a graphical note explaining what the different colored amino acids represent. See page 10 lines 275-281.
- line 327 What does it mean “realm of allergenicity”? Please correct it.
Answer: Thanks very much for your comments. We have corrected that sentence. See page 12 lines 338-339.
- line 333 What does it mean “cation city”? Please correct it.
Answer: Thank you very much for your comment. We corrected "cation city" to "cation activity". See page 12 line 345.
- line 396 I suggest “both these bacterial species” instead of “two bacterial species”.
Answer: Thank you very much for your comment. Based on your suggestion, we have replaced the original text. See page 13 line 411.

Reviewer 3 Report
Comments and Suggestions for Authors
The authors satisfactorily complied with all suggestions. No more comments.
Author Response
Response to Reviewers
Dear Editor and Reviewers,
Thank you for your great interest in our first draft and for your approval, which gives us great confidence.
